# Children's emotion inferences from masked faces: Implications for social interactions during COVID-19

**Ashley L. Ruba**ⓘ*, **Seth D. Pollak**

Department of Psychology and Waisman Center, University of Wisconsin – Madison, Madison, Wisconsin, United States of America

* ruba@wisc.edu

## Abstract

To slow the progression of COVID-19, the Centers for Disease Control (CDC) and the World Health Organization (WHO) have recommended wearing face coverings. However, very little is known about how occluding parts of the face might impact the emotion inferences that children make during social interactions. The current study recruited a racially diverse sample of school-aged (7- to 13-years) children from publicly funded after-school programs. Children made inferences from facial configurations that were not covered, wearing sunglasses to occlude the eyes, or wearing surgical masks to occlude the mouth. Children were still able to make accurate inferences about emotions, even when parts of the faces were covered. These data suggest that while there may be some challenges for children incurred by others wearing masks, in combination with other contextual cues, masks are unlikely to dramatically impair children's social interactions in their everyday lives.

## Introduction

COVID-19 is one of the worst pandemics in modern history. To slow the spread of the virus, both the Centers for Disease Control and the World Health Organization have recommended wearing face coverings in public spaces. This recommendation has led to speculation and concern about the ramifications of mask wearing on emotion communication [1, 2]. Of particular concern for parents and teachers is how wearing masks might impact children's social interactions [3, 4]. While much research has documented how children infer emotions from facial configurations and how this ability predicts children's social and academic competence [5–7], uncertain is how children make these inferences when part of the face is occluded by a mask. The current study explores how children draw emotional inferences from faces partially occluded by surgical masks and, as a comparison, sunglasses.

Paper and cloth "surgical" masks cover the lower half of the face, allowing the eyes, eyebrows, and forehead to remain visible. When asked to infer emotions from stereotypical facial configurations, adults tend to look at the eyes first and more frequently than other facial features (e.g., mouth, nose) [8–10], although scan patterns vary across cultures [11, 12]. Adults

**Data Availability Statement:** The raw data and analysis code are available from the Open Science Framework (doi: 10.17605/OSF.IO/7FYX9).

**Funding:** SDP was supported by the National Institute of Mental Health (R01 MH61285) and a

core grant to the Waisman Center from the National Institute of Child Health and Human Development (U54 HD090256). ALR was supported by an Emotion Research Training Grant (T32 MH018931) from the National Institute of Mental Health. Funders did not play any role in study design, data collection and analysis, decision to publish, or preparation of the manuscript.

**Competing interests:** The authors have declared that no competing interests exist.

also show above-chance accuracy at inferring emotions from stereotypical facial configurations when only the eyes are visible [13–15]. (Note: Across most studies, "accuracy" refers to whether participants select the label/emotion typically associated with a particular facial configuration, such as "anger" for a face with furrowed brows and a tight mouth). Humans are particularly sensitive to eyes [16], and thus, eye musculature may convey sufficient information for adults to make reasonably accurate emotional inferences, even when masks cover the mouth and nose.

Nevertheless, focusing on the eyes alone may be insufficient for some emotion inferences [11, 17]. When facial configurations are ambiguous or subtle, adults (and children) shift their attention between the eyes and other facial features that may provide additional diagnostic information [18]. For instance, to make inferences about whether wide eyes indicate "fear" or "surprise," adults attend to both the eyes and the mouth [19, 20]. Adults also tend to fixate on specific facial features that characterize specific emotion stereotypes, such as the mouth for happiness and the nose for disgust [8–10, 21]. Inferring emotions from these characteristic facial features (e.g., labeling a smile as "happy") is also influenced by other parts of the face [10, 22–24]. In short, adults scan facial configurations in a holistic manner [19, 25], allowing for information to be gleaned from the mouth, nose, and other parts of the face, which are not accessible when wearing a mask.

While this research suggests how adults infer emotions when parts of the face are obscured, much less is known about how this process emerges in early childhood [7]. In the first year of life, infants shift from configural to holistic processing of faces [26, 27] and demonstrate heightened attention to eyes associated with positive affective states [28–30]. By 3-years of age, children show above-chance accuracy at inferring emotions from the eyes alone [31]. However, compared to when other parts of the face are also visible, 5- to 10-year-olds are less accurate at inferring emotions from the eyes only [32–35], although results for specific emotions have been inconsistent across studies [32, 33]. One study even found that 3- to 4-year-olds were *more accurate* at inferring happiness, sadness, and surprise from faces when the eyes were *covered* by sunglasses [35]. With respect to emotion inferences with masks, only one study has obscured the mouth (with a dark circle). Roberson et al. (2012) found that 9- to 10-year-old children and adults showed more accurate emotion inferences for uncovered faces than when the mouth was covered. However, 3- to 8-year-olds did not show these impairments. Similarly, when facial configurations are presented within a background emotion context, 12-year-olds show heightened visual attention to faces compared to 4- and 8-year-olds [18]. Thus, there may be developmental differences in children's reliance on and use of specific facial features to make emotional inferences and the impact of mask wearing on these inferences.

## Current study

The current study examines how 7- to 13-year-old children draw emotional inferences from facial configurations that are partially occluded. This age range was selected because there is a shift during this time in children's use of eye information to infer others' emotions [33, 35]. Facial configurations associated with different negative emotions (i.e., sadness, anger, fear) were presented via a Random Image Structure Evolution (RISE) paradigm [36]. Facial configurations were initially presented in a highly degraded format in which children only had access to partial facial information. In a dynamic sequence, the images became less degraded at regular intervals. After each interval, children selected from an array of emotion labels to indicate their belief about how the person displaying the facial configuration was feeling. Thus, this paradigm allows for the assessment of children's emotion inferences from incomplete through

more complete facial information. This approach is more similar to daily experiences with the unfolding of others' emotions compared to a single presentation of a facial configuration at full intensity [37].

We examined how children perceived others' emotions as partial information about the face was presented, to evaluate whether masks meaningfully changed the types of inferences children made. We included sunglasses as a comparison for other types of coverings that children regularly encounter on faces in their daily lives. Together, these results shed light on how mask wearing during COVID-19 might—or might not—influence children's inferences about others' emotions and their related social interactions.

## Methods

### Participants

Procedures were approved by the University of Wisconsin—Madison Institutional Review Board. To test a racially diverse sample of children, participants were recruited from publicly funded after-school programs associated with the Dane County (Wisconsin) Department of Human Services. The final sample included 81 7- to 13-year-old children (37 female, $M$ = 9.86 years, $SD$ = 1.84 years, range = 7.06–12.98 years). Parents identified their children as Black (53%, $n$ = 43), White (41%, $n$ = 33), and Multi-racial (6%, $n$ = 5). Three additional children participated in the study but were excluded from final analyses due to missing or corrupt data files. A power analysis confirmed that this sample size would be sufficient to detect reliable differences in a within-subjects design, assuming a medium effect size ($f$ = .25) at the .05 level [36].

### Stimuli

Stimuli, selected from the Matsumoto and Ekman (1988) database, were pictures of stereotypical facial configurations associated with sadness, anger, and fear posed by male and female models. These three emotions were selected given that adults tend to fixate predominantly on the eyes for these facial configurations, rather than other parts of the face (e.g., the mouth and nose, as with happiness and disgust) [8, 20]. Further, negative emotions are complex and rich in informational value [38]; yet, these emotions have received limited empirical attention in the literature on emotion perception development [39–41]. Pictures were presented in unaltered format (i.e., with no covering) or digitally altered to be (a) covered with a surgical face mask that obscured the mouth and nose, or (b) covered with sunglasses that obscured the eyes and eyebrows (see Fig 1). Pictures of each emotion (sad, anger, fear) paired with each covering type (none, mask, shades) were presented twice in a random order (i.e., 18 stimuli total). Half of the presentations were on male faces and half were on female faces.

### Procedure

Parents provided written consent and children provided verbal assent prior to participation. Children were tested in a modified Random Image Structure Evolution (RISE) paradigm [36]. RISE performs pairwise exchanges of pixels in an image until the target image dissolves into an unstructured random field. These exchanges are presented in reverse order such that participants begin viewing a random visual display that gradually transforms into a fully formed, clear image (see Fig 2). Importantly, the RISE protocol holds the low-level perceptual attributes of the original image (e.g., luminance, color) constant.

Children viewed these image sequences on a high-resolution touch-sensitive color monitor. Faces were initially presented in a highly degraded format. At 14 regular 3.3-s intervals, the images became less degraded and easier to discern. After each interval, children were

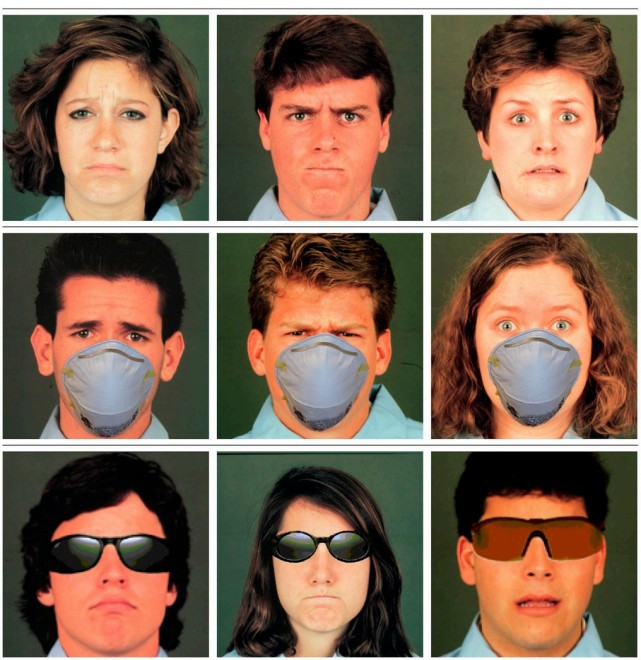

**Fig 1. Example stimuli by covering.** From top to bottom: none, mask, shades. From left to right: sad, anger, fear. The image is from a set of photographs entitled *Japanese and Caucasian Facial Expressions of Emotion (JACFEE)* by D. Matsumoto and P. Ekman, University of California, San Francisco, 1988. Copyright 1988 by D. Matsumoto and P. Ekman. Reprinted by permission.

prompted to identify the emotion depicted on the face by selecting one of the following emotion labels: "happy," "sad," "angry," "surprised," "afraid," or "disgusted." Labels were presented in this order on the screen, and children touched a label to indicate their response. A total of 252 responses were collected for each child (i.e., 14 trials each of 18 stimuli). Responses were coded as "accurate" if the child selected the label/emotion typically associated with a particular facial configuration (i.e., "anger" for a face with furrowed brows).

## Results

All analyses were conducted in *R* [42], and figures were produced using the package *ggplot2* [43]. Alpha was set at $p < .05$. The raw data and analysis code are available on OSF: doi.org/10.17605/OSF.IO/7FYX9. Children's accuracy scores were analyzed in a 3 (Emotion: sad / anger / fear) x 3 (Covering: none / mask / shades) x 14 (Trial: 1 to 14) repeated-measures ANCOVA with child Gender as a between-subjects factor and child Age as a covariate. All significant main effects and interactions are explored below.

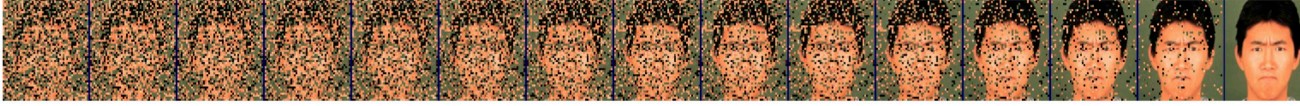

**Fig 2. Example test sequence.** Anger (no covering) is pictured. The image is from a set of photographs entitled *Japanese and Caucasian Facial Expressions of Emotion (JACFEE)* by D. Matsumoto and P. Ekman, University of California, San Francisco, 1988. Copyright 1988 by D. Matsumoto and P. Ekman. Reprinted by permission.

### Are children more accurate with anger, sadness, or fear?

The main effect of Emotion, $F(2, 154) = 30.46$, $p < .001$, $\eta_p^2 = .28$, showed that children were more accurate with facial configurations associated with sadness ($M = .36$, $SD = .48$) compared to anger ($M = .27$, $SD = .44$), $t(80) = 4.10$, $p < .001$, $d = .46$, $CI_{95\%}[.04, .13]$, or fear ($M = .19$, $SD = .39$), $t(80) = 7.39$, $p < .001$, $d = .82$, $CI_{95\%}[.12, .21]$. Children were also more accurate with facial configurations associated with anger compared to fear, $t(80) = 3.68$, $p < .001$, $d = .41$, $CI_{95\%}[.04, .12]$.

A similar pattern of results was seen in the Emotion x Trial interaction, $F(14, 1091) = 5.35$, $p < .001$, $\eta_p^2 = .06$, which was explored with 95% confidence intervals (estimated with bootstrapping, Fig 3). Unsurprisingly, children became more accurate with each emotion as the images became less obscured. In earlier trials, there were few differences between the stimuli. In later trials, children were more accurate with facial configurations associated with sadness compared to anger and fear, and children were more accurate with facial configurations associated with anger compared to fear.

### Are children less accurate with the eyes or mouth covered?

The primary question addressed by this study is whether masks meaningfully degraded children's ability to infer others' emotions. The main effect of Covering, $F(2, 154) = 27.19$ $p < .001$, $\eta_p^2 = .26$, showed that children were more accurate when faces were uncovered ($M = .34$, $SD = .47$) compared to when the faces wore a mask ($M = .24$, $SD = .43$), $t(80) = 6.57$, $p < .001$, $d = .73$, $CI_{95\%}[.07, .13]$, or shades ($M = .24$, $SD = .43$), $t(80) = 6.24$, $p < .001$, $d = .69$, $CI_{95\%}[.07, .13]$. Accuracy between the faces that wore masks and shades did not differ, $t(80) = .20$, $p >$

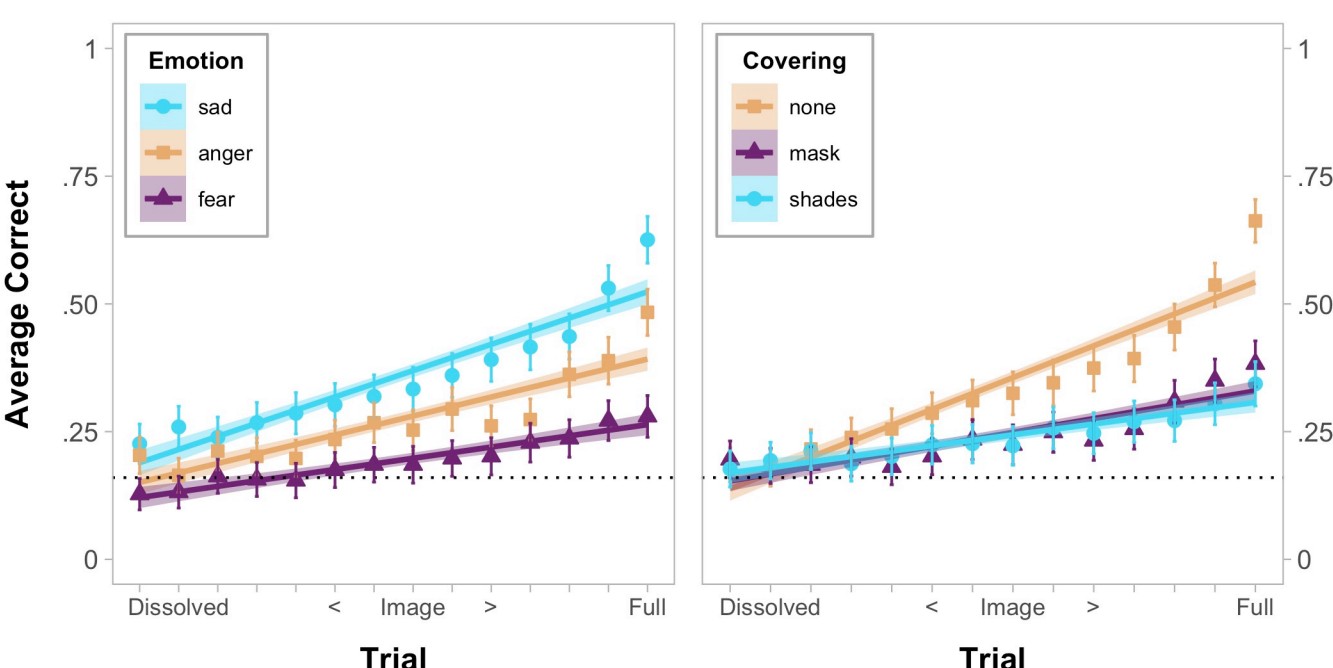

**Fig 3. Linear regression and means for emotion x trial and covering x trial interactions.** The dotted line indicates chance responding (1/6). Confidence intervals (95%) were estimated with bootstrapping (1,000 bootstrap estimates resampled 81 times from mean participant accuracy).

.25, $d$ = .02, CI$_{95\%}$[-.03, .03]. A similar pattern of results was seen in the Covering x Trial interaction, $F(18, 1372)$ = 10.27, $p$ < .001, $\eta_p^2$ = .12, which was also explored with 95% confidence intervals (estimated with bootstrapping, Fig 3). Yet, the overall effect of face coverings on accuracy was relatively small, especially as children gained more visual information.

## How do different coverings impact children's inferences for specific emotions?

To explore the Emotion x Covering interaction, $F(4, 284)$ = 3.58, $p$ = .009, $\eta_p^2$ = .04, paired t-tests were conducted between each covering type, separated by emotion (Fig 4). Further, to examine if children's performance was greater than chance (m = 1/6) for each emotion-covering pair, additional one-sample t-tests were conducted. Bonferroni-holm corrections were applied for multiple comparisons (reported p-values are corrected).

For facial configurations associated with sadness, children were less accurate when the faces wore masks ($M$ = .28, $SD$ = .45) compared to when the faces had no covering ($M$ = .43, $SD$ = .49), $t(80)$ = 4.60, $p$ < .001, $d$ = .51, CI$_{95\%}$[.08, .21]. Children's accuracy did not differ when the faces wore shades ($M$ = .37, $SD$ = .48) compared to when the faces had no covering, $t(80)$ = 1.91, $p$ = .12, $d$ = .21, CI$_{95\%}$[.00, .12], or wore masks $t(80)$ = 2.47, $p$ = .063, $d$ = .27, CI$_{95\%}$[.02, .16]. Children responded with above-chance accuracy for all coverings: none, $t(80)$ = 10.30, $p$ < .001, $d$ = 1.14, CI$_{95\%}$[.38, .47]; mask, $t(80)$ = 4.77, $p$ < .001, $d$ = .53, CI$_{95\%}$[.23, .33], shades, $t(80)$ = 7.23, $p$ < .001, $d$ = .80, CI$_{95\%}$[.31, .42].

For facial configurations associated with anger, children were less accurate when the faces wore masks ($M$ = .27, $SD$ = .44) compared to when the faces had no covering ($M$ = .34, $SD$ = .48), $t(80)$ = 2.72, $p$ = .041, $d$ = .30, CI$_{95\%}$[.02, .13]. Children were also less accurate when the faces wore shades ($M$ = .20, $SD$ = .40) compared to when the faces had no covering, $t(80)$ = 5.01, $p$ < .001, $d$ = .56, CI$_{95\%}$[.09, .20]. Children's accuracy when the faces wore masks or shades did not differ, $t(80)$ = 2.16, $p$ = .10, $d$ = .24, CI$_{95\%}$[.01, .13]. Children only responded

## Emotion x Covering

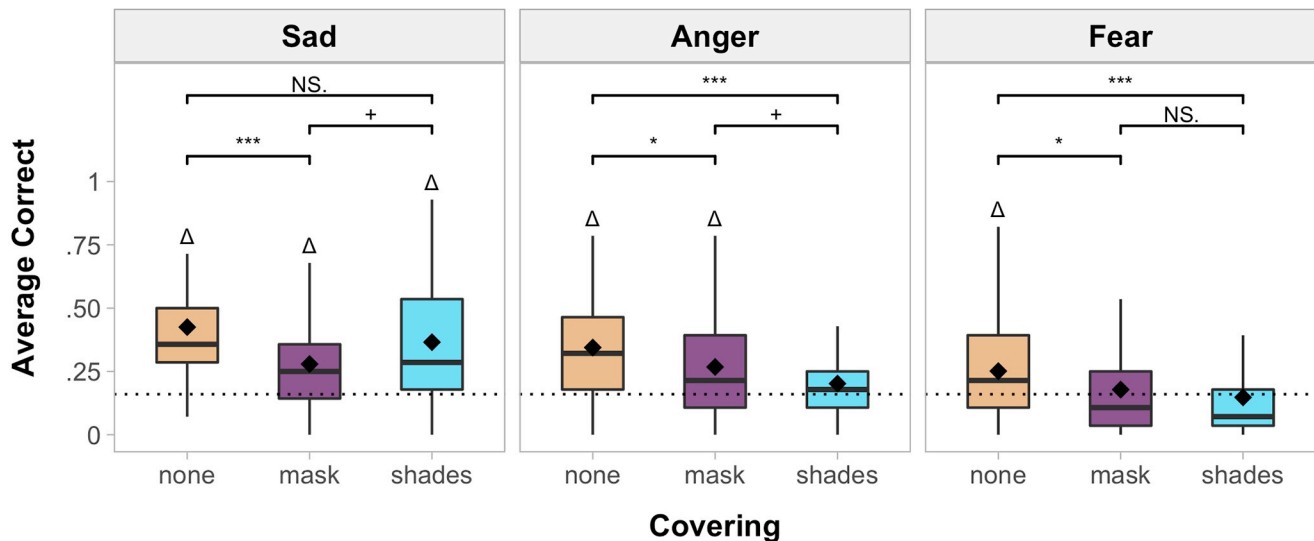

**Fig 4. Boxplots for the emotion x covering interaction.** * indicates comparisons between covering types for each emotion ($^*p$ < .05, $^{***}p$ < .001, $^+p$ < .10, NS = not significant with Bonferroni-Holm corrections). The dotted line indicates chance responding (1/6). Δ indicates that accuracy was significantly greater than chance ($p$ < .05 with Bonferroni-Holm corrections).

with above-chance accuracy when the faces had no covering, $t(80) = 7.28$, $p < .001$, $d = .81$, $CI_{95\%}[.30, .39]$, or when the faces wore masks, $t(80) = 4.50$, $p < .001$, $d = .50$, $CI_{95\%}[.22, .31]$. Children did not respond with above-chance accuracy when the faces wore shades, $t(80) = 1.77$, $p = .24$, $d = .20$, $CI_{95\%}[.16, .24]$.

For facial configurations associated with fear, children were less accurate when the faces wore masks ($M = .18$, $SD = .38$) compared to when the faces had no covering ($M = .25$, $SD = .43$), $t(80) = 2.91$, $p = .028$, $d = .32$, $CI_{95\%}[.02, .12]$. Children were also less accurate when the faces wore shades ($M = .15$, $SD = .35$) compared to when the faces had no covering, $t(80) = 3.96$, $p < .001$, $d = .44$, $CI_{95\%}[.05, .16]$. Children's accuracy when the faces wore masks or shades did not differ, $t(80) = 1.09$, $p > .25$, $d = .12$, $CI_{95\%}[-.02, .09]$. Children only responded with above-chance accuracy when the faces had no covering, $t(80) = 3.85$, $p < .001$, $d = .43$, $CI_{95\%}[.21, .30]$. Children did not reach above-chance accuracy when the faces wore masks, $t(80) = .50$, $p > .25$, $d = .06$, $CI_{95\%}[.13, .22]$, or shades, $t(80) = .94$, $p > .25$, $d = .10$, $CI_{95\%}[.11, .19]$.

Thus, across all emotions, children were less accurate with faces that wore a mask compared to faces that were not covered. However, children were only less accurate with faces that wore sunglasses compared to uncovered for two emotions: anger and fear. This suggests that children inferred whether the face displayed sadness from mouth shape alone, whereas the information from the eye region was necessary for forming inferences about anger and fear (see below). Ultimately, accuracy differences between the masks and shades did not significantly differ for any emotion. Thus, while both types of coverings negatively impacted children's emotion inferences, the strongest impairments were observed for facial configurations associated with fear.

**What inferences did children make for each stimulus?.** To further investigate why children did not reach above-chance responding for the *anger-shades*, *fear-mask*, and *fear-shades* stimuli, we examined children's responses to each stimulus. As seen in Fig 5, children tended to interpret facial configurations associated with fear as "surprised." This effect was particularly pronounced when the faces were covered by a mask. Children also tended to interpret facial configurations associated with anger as "sad" when the faces were covered by shades. In contrast, children interpreted facial configurations associated with sadness as "sad," regardless of covering.

## How does children's accuracy differ based on age?

The main effect of Age, $F(1, 78) = 5.85$, $p = .018$, $\eta_p^2 = .07$, showed that accuracy improved as child age increased. The Age x Trial, $F(6, 474) = 2.40$, $p = .027$, $\eta_p^2 = .03$, interaction was explored with a simple slopes analysis. This analysis revealed that older children showed enhanced performance over the course of the experiment compared to younger children (Fig 6).

## How does children's accuracy differ based on gender?

Although there was not a significant main effect of Gender, $F(1, 78) = .54$, $p > .25$, $\eta_p^2 = .01$, a Gender x Emotion interaction emerged, $F(2, 154) = 3.20$, $p = .044$, $\eta_p^2 = .04$. Follow-up comparisons showed that male participants were significantly more accurate with facial configurations associated with anger ($M = .30$, $SD = .46$) compared to female participants ($M = .24$, $SD = .42$), $t(79) = 2.28$, $p = .025$, $d = .51$, $CI_{95\%}[.01, .12]$. Accuracy for facial configurations associated with sadness, $t(79) = 1.25$, $p = .22$ $d = .28$, $CI_{95\%}[-.03, .11]$, or fear, $t(79) = .53$, $p > .25$, $d = .12$, $CI_{95\%}[-.08, .05]$, did not differ based on participant gender.

## Discussion

These results highlight how children's social interactions may be minimally impacted by mask wearing during the COVID-19 pandemic. Positive social interactions are predicated on the

## Responses by Emotion and Covering

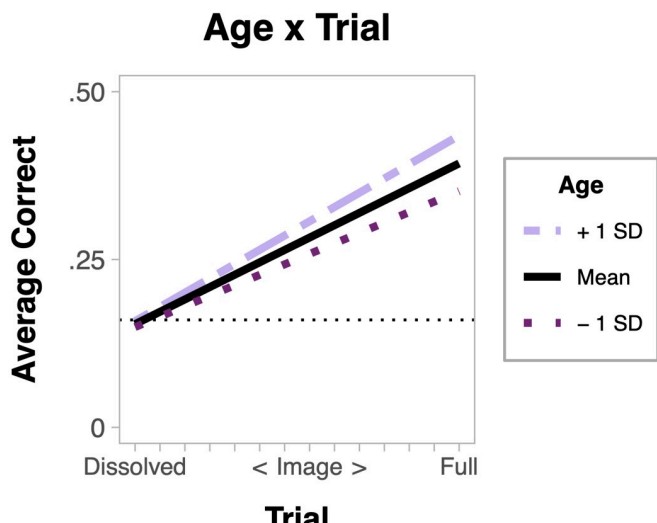

**Fig 5. Average frequency of responses for each emotion and covering as a function of trial.** The expected ("accurate") response for each stimulus is outlined in white.

## Age x Trial

**Fig 6. Simple slopes analysis for age x trial interaction.** The dotted line indicates chance responding (1/6).

ability to accurately infer and respond to others' emotions. In the current study, children's emotion inferences about faces that wore masks compared to when faces were not covered were still above chance. Masks seem to have the greatest effect on children's inferences about facial configurations associated with "fear," which were commonly identified as "surprised" when the mouth and nose were covered. Thus, although children may require more visual facial information to infer emotions with masks, children may reasonably infer whether someone wearing a mask is sad or angry, based on the eye region alone. In addition, children's accuracy with masked facial configurations did not significantly differ from their accuracy with facial configurations that wore sunglasses—a common accessory that children encounter in their everyday lives. Thus, it appears that masks do not negatively impact children's emotional inferences to a greater degree than sunglasses. In sum, children's ability to infer and respond to another person's emotion, and their resulting social interactions, may not be dramatically impaired by mask wearing during the COVID-19 pandemic.

Furthermore, in everyday life, it is unlikely that children draw emotional inferences from facial configurations alone. For instance, the same facial configuration may be inferred as either "anger" or "disgust" depending on background context, body posture, and facial coloration [18, 44, 45]. In addition, dynamic facial configurations and faces that are vocalizing are scanned differently than silent, static pictures of faces [46–48]. Ultimately, facial configurations displayed in everyday life are more dependent on context, less consistent, and less specific than pictures of stereotyped emotions commonly used in laboratory tasks [37]. The current paradigm improves upon these standard laboratory tasks by assessing children's emotion inferences from incomplete facial information. However, the key to children's emotional inferences is the ability to learn about and navigate the tremendous variability inherent in human emotion [7, 49]. In everyday life, children may be able to use additional contextual cues to make reasonably accurate inferences about others' variable emotional cues, even if others are wearing masks.

Future research should take these considerations into account when designing and interpreting findings on mask wearing during the COVID-19 pandemic. While the current study assessed whether children made "accurate" emotion inferences, a single facial configuration can be interpreted in many ways that are "accurate" given a particular context [37]. Researchers could explore how children make emotion inferences from a wider variety of non-stereotyped emotional cues that are presented in context. Although we did not find many age effects in the current study, future research could also explore how younger children's social interactions are impacted by mask wearing, particularly infants who are actively learning about others' emotions [50]. To conclude, while there may be some loss of emotional information due to mask wearing, children can still infer emotions from faces, and likely use many other cues to make these inferences. This suggests that children, and adults, may be able to adapt to the new reality of mask wearing to have successful interactions during this unprecedented health crisis.

## Author Contributions

**Conceptualization:** Seth D. Pollak.

**Formal analysis:** Ashley L. Ruba.

**Funding acquisition:** Seth D. Pollak.

**Investigation:** Seth D. Pollak.

**Methodology:** Seth D. Pollak.

**Project administration:** Seth D. Pollak.

**Resources:** Seth D. Pollak.

**Visualization:** Ashley L. Ruba.

**Writing – original draft:** Ashley L. Ruba.

**Writing – review & editing:** Ashley L. Ruba, Seth D. Pollak.

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
