## [Decision Letter · Decision Letter 0]

9 Nov 2020

PONE-D-20-30132

Children’s Emotion Inferences from Masked Faces: Implications for Social Interactions During COVID-19

PLOS ONE

Dear Dr. Ruba,

Thank you for submitting your manuscript to PLOS ONE. After careful consideration, we feel that it has merit but does not fully meet PLOS ONE’s publication criteria as it currently stands. Therefore, we invite you to submit a revised version of the manuscript that addresses the points raised during the review process.

We look forward to receiving your revised manuscript.

Kind regards,

Zezhi Li, Ph.D., M.D.

Academic Editor

PLOS ONE

Journal Requirements:

3. Please ensure that you refer to Figure 4 in your text as, if accepted, production will need this reference to link the reader to the figure.

4. We note that Figures 1 and 2 include images of participants in the study. 

Reviewers' comments:

Reviewer's Responses to Questions

**Comments to the Author**

1. Is the manuscript technically sound, and do the data support the conclusions?

Reviewer #1: Yes

Reviewer #2: Yes

2. Has the statistical analysis been performed appropriately and rigorously? 

Reviewer #1: Yes

Reviewer #2: Yes

3. Have the authors made all data underlying the findings in their manuscript fully available?

Reviewer #1: Yes

Reviewer #2: Yes

4. Is the manuscript presented in an intelligible fashion and written in standard English?

Reviewer #1: Yes

Reviewer #2: Yes

5. Review Comments to the Author

Reviewer #1: Thanks to the editor for giving me this opportunity to review this article. Overall, this article is good, and it is also a useful exploration in a special period. There is one main question that I think it still needs to be adjusted statistically. The author performed repeated measures analysis of variance and took age as a covariate. In fact, gender is also a very important factor that may influence the results, so I suggest that the author stratify gender and see what the results are different?

Reviewer #2: The objective of the research is to assess how occluding parts of the face might impact the emotion inferences that children make during social interactions. The study found that while there may be some challenges for children incurred by others wearing masks, in combination with other contextual cues, masks are unlikely to dramatically impair children’s social interactions in their everyday lives.

Although this study is quite interesting and these results highlight how children’s social interactions may be minimally impacted by mask wearing during the COVID-19 pandemic, the following issues need to be addressed before it is accepted and published.

Points to address:

1. The current study mainly focused on negative emotions i.e. sadness, anger, and fear. I wonder is there a particular reason why only study these 3 negative emotions? It would be great if the authors can provide the motivation for studying these negative emotions in the introduction.

2. Can you describe how the 1000 bootstrapped samples were estimated in detail?

3. How does children’s accuracy differ based on gender? It would be great if the results can be included in the manuscript.

4. For facial configurations associated with sadness, it seems there is no significant difference in accuracy between no covering and shades. How do you interpret this finding?

6. PLOS authors have the option to publish the peer review history of their article (what does this mean?). If published, this will include your full peer review and any attached files.

Reviewer #1: No

Reviewer #2: No

---

## [Author Response · Author response to Decision Letter 0]

17 Nov 2020

Dear Dr. Li,

Thank you for providing us with this opportunity to revise and resubmit our work. We sincerely appreciate the reviewers’ thoughtful and thorough comments. Please find attached our revised manuscript to be reconsidered for publication at PLOS One. 

As indicated below, this revised manuscript takes into account all of the issues raised by you and the reviewers. We have included both an unmarked and marked-up version of our manuscript in this submission.

Editor Comments:

We have modified our files to meet PLOS ONE’s style requirements, including those for file naming.

The data and code can be located at this DOI: doi.org/10.17605/OSF.IO/7FYX9. This link has been added to the manuscript (page 8). The Data Availability statement has also been updated. 

3. Please ensure that you refer to Figure 4 in your text as, if accepted, production will need this reference to link the reader to the figure.

Figure 4 has now been referenced on page 9.

4. We note that Figures 1 and 2 include images of participants in the study. Please amend the methods section and ethics statement of the manuscript to explicitly state that the patient/participant has provided consent for publication: “The individual in this manuscript has given written informed consent (as outlined in PLOS consent form) to publish these case details”.

Figures 1 and 2 are pictures of stimuli used in the study (selected from the Matsumoto and Ekman (1988) database), not pictures of participants in the study. We have now explicitly stated this in the captions for both figures (pages 6-7)

Reviewer #1: 

1. The author performed repeated measures analysis of variance and took age as a covariate. In fact, gender is also a very important factor that may influence the results, so I suggest that the author stratify gender and see what the results are different?

We have added Gender as a between-subjects factor to the overall ANCOVA and reported results on page 12:

“Although there was not a significant main effect of Gender, F(1, 78) = .54, p > .25, ηp2 = .01, a Gender x Emotion interaction emerged, F(2, 154) = 3.20, p = .044, ηp2 = .04. Follow-up comparisons showed that male participants were significantly more accurate with facial configurations associated with anger (M = .30, SD = .46) compared to female participants (M = .24, SD = .42), t(79) = 2.28, p = .025, d = .51, CI95%[.01, .12]. Accuracy for facial configurations associated with sadness, t(79) = 1.25, p = .22 d = .28, CI95%[-.03, .11], or fear, t(79) = .53, p > .25, d = .12, CI95%[-.08, .05], did not differ based on participant gender.”

Reviewer #2:

1. I wonder is there a particular reason why only study these 3 negative emotions? It would be great if the authors can provide the motivation for studying these negative emotions in the introduction.

We have clarified why these three negative emotions were selected on page 7:

“These three emotions were selected given that adults tend to fixate predominantly on the eyes for these facial configurations, rather than other parts of the face (e.g., the mouth and nose, as with happiness and disgust) (Eisenbarth & Alpers, 2011; Smith et al., 2005). Further, negative emotions are complex and rich in informational value (Vaish et al., 2008); yet, these emotions have received limited empirical attention in the literature on emotion perception development (Ruba et al., 2019, 2020; Ruba & Repacholi, 2019).”

2. Can you describe how the 1000 bootstrapped samples were estimated in detail?

We have provided more detail on the bootstrapped samples in the caption of Figure 3 (page 8):

“Confidence intervals (95%) were estimated with bootstrapping (1,000 bootstrap estimates resampled 81 times from mean participant accuracy).”

3. How does children’s accuracy differ based on gender? It would be great if the results can be included in the manuscript.

We have added Gender as a between-subjects factor to the overall ANCOVA and reported results on page 12:

“Although there was not a significant main effect of Gender, F(1, 78) = .54, p > .25, ηp2 = .01, a Gender x Emotion interaction emerged, F(2, 154) = 3.20, p = .044, ηp2 = .04. Follow-up comparisons showed that male participants were significantly more accurate with facial configurations associated with anger (M = .30, SD = .46) compared to female participants (M = .24, SD = .42), t(79) = 2.28, p = .025, d = .51, CI95%[.01, .12]. Accuracy for facial configurations associated with sadness, t(79) = 1.25, p = .22 d = .28, CI95%[-.03, .11], or fear, t(79) = .53, p > .25, d = .12, CI95%[-.08, .05], did not differ based on participant gender.”

4. For facial configurations associated with sadness, it seems there is no significant difference in accuracy between no covering and shades. How do you interpret this finding?

We have added an interpretation to the revision (page 11). Our analysis in the subsequent section (“what inferences did children make for each stimulus”) also provides support for this interpretation.

“However, children were only less accurate with faces that wore sunglasses compared to uncovered for two emotions: anger and fear. This suggests that children inferred whether the face displayed sadness from mouth shape alone, whereas the information from the eye region was necessary for forming inferences about anger and fear.”

---

## [Decision Letter · Decision Letter 1]

30 Nov 2020

Children’s Emotion Inferences from Masked Faces: Implications for Social Interactions During COVID-19

PONE-D-20-30132R1

Dear Dr. Ruba,

We’re pleased to inform you that your manuscript has been judged scientifically suitable for publication and will be formally accepted for publication once it meets all outstanding technical requirements.

Kind regards,

Zezhi Li, Ph.D., M.D.

Academic Editor

PLOS ONE

Additional Editor Comments (optional):

Reviewers' comments:

Reviewer's Responses to Questions

**Comments to the Author**

1. If the authors have adequately addressed your comments raised in a previous round of review and you feel that this manuscript is now acceptable for publication, you may indicate that here to bypass the “Comments to the Author” section, enter your conflict of interest statement in the “Confidential to Editor” section, and submit your "Accept" recommendation.

Reviewer #2: All comments have been addressed

2. Is the manuscript technically sound, and do the data support the conclusions?

Reviewer #2: Yes

3. Has the statistical analysis been performed appropriately and rigorously? 

Reviewer #2: Yes

4. Have the authors made all data underlying the findings in their manuscript fully available?

Reviewer #2: Yes

5. Is the manuscript presented in an intelligible fashion and written in standard English?

Reviewer #2: Yes

6. Review Comments to the Author

Reviewer #2: The authors have satisfactorily responded to all my questions and made the necessary changes to the manuscript. The revised version of the manuscript appears to be good.

7. PLOS authors have the option to publish the peer review history of their article (what does this mean?). If published, this will include your full peer review and any attached files.

Reviewer #2: No

---

## [Editor Report · Acceptance letter]

4 Dec 2020

PONE-D-20-30132R1 

Children’s Emotion Inferences from Masked Faces: Implications for Social Interactions During COVID-19 

Dear Dr. Ruba:

I'm pleased to inform you that your manuscript has been deemed suitable for publication in PLOS ONE. Congratulations! Your manuscript is now with our production department. 

Kind regards, 

on behalf of

Dr. Zezhi Li 

Academic Editor

PLOS ONE